# A qualitative study on knowledge, perception, and practice related to non-communicable diseases in relation to happiness among rural and urban residents in Bhutan

**Hiromi Kohori-Segawa** [1]*, **Chencho Dorji** [2], **Kunzang Dorji**[2], **Ugyen Wangdi**[2], **Chimi Dema**[3], **Yankha Dorji**[3], **Patou Masika Musumari**[4], **Teeranee Techasrivichien**[4], **Sonia Pilar Sugimoto Watanabe**[4], **Ryota Sakamoto**[5], **Masako Ono-Kihara**[4], **Masahiro Kihara**[4], **Yuichi Imanaka**[1]

1 Department of Healthcare Economics and Quality Management, Graduate School of Medicine, Kyoto University, Kyoto, Japan, 2 Faculty of Nursing and Public Health, Khesar Gyalpo University of Medical Sciences of Bhutan, Thimphu, Bhutan, 3 Ministry of Health, Phuntholing, Chukha, Bhutan, 4 Department of Global Health and Socio-epidemiology, Graduate School of Medicine, Kyoto University, Kyoto, Japan, 5 Department of Environmental Coexistence, Center for Southeast Asian Studies, Kyoto University, Kyoto, Japan

* segawa.hiromi.57r@kyoto-u.jp

**Data Availability Statement:** This study was conducted in the Kingdom of Bhutan which, is a small Himalayan country. The transcript,

## Abstract

### Purpose

Bhutan, known as a country of happiness, has experienced rapid social changes and the increasing burden of non-communicable diseases (NCDs) that can impact health and happiness. To inform future NCD prevention programs in Bhutan, this study explores knowledge, perception, and the practices of Bhutanese related to NCDs in the context of the philosophy of happiness.

### Methodology

Research was conducted in rural and urban communities of Bhutan in 2017 among 79 inhabitants of both genders, aged ≥18. Participants were recruited through purposive sampling with the data collected by in-depth interviews, participatory observation, and anthropometric measurements. Data were analyzed by thematic analysis.

### Results/Discussion

Across participants, health was considered as an important element of "happiness". However, lifestyle-related NCD risk factors prevailed due to the lack of effective education programs on NCDs and thus the lack of practical knowledge for NCD prevention across society. We further found that the value of happiness "finding happiness in any situation is virtue" was universal as well as other traditional values and customs, shaping people's health behaviors. From these observations, it is recommended that more practical NCD

anthropometric, and participatory observation data and photos, contain participant information including disease history, relationships with others, religion, family history, social status, and sensitive issues, such as family problems and mental health. Given these factors, the data cannot be made publicly available considering the potential breach of privacy and confidentiality. However, to prove the validity of coding, we have shown a part of interview or observation data in Table 4 to Table 9 in this manuscript. And the authors can share the data if requested through the Research Ethics Board of Health at the Royal Government of Bhutan, and the ethics committee of Kyoto University. Contact information: The Research Ethics Board of Health, Royal Government of Bhutan, PABX: + 975-2-322602, 322351, 328091, 328092, 328093 (Extension 333) Fax: 324649 (Research No. 2017-027) Ethics Committee of Kyoto University Graduate School and Faculty of Medicine, Kyoto University Hospital, E-mail: ethcom@kuhp.kyoto-u.ac.jp (Research Number: No. R1059)

**Funding:** 1) Hiromi Kohori-Segawa: MEXT, leading graduate school for Sustainable Development and Global Survivability Studies, centre for educational program promotion in graduate school, Kyoto university: Grant Number N/A: http://www.gss. kyoto-u.ac.jp/english/ 2) Hiromi Kohori-Segawa: Super Global Course in School of Public Health, Kyoto university: Grant Number N/A: http://sph. med.kyoto-u.ac.jp/en/info/international-programs/ 3) Hiromi Kohori-Segawa: WENDI, leading graduate school for Sustainable Development and Global Survivability Studies, centre for educational program promotion in graduate school, Kyoto university: Grant Number N/A: http://www.gss. kyoto-u.ac.jp/english/ 4) Yuichi Imanka: JSPS KAKENHI:Grant Number JP19H01075: https:// www.jsps.go.jp/j-grantsinaid/01_seido/01_ shumoku/index.html The funders had no role in study design, data collection and analysis, decision to publish, or preparation of the manuscript.

**Competing interests:** The authors have declared that no competing interests exist.

education/prevention programs should be urgently introduced in Bhutan that involve multiple generations, religion authorities, educational settings, and medical services.

## Originality

This is the first comprehensive qualitative study on the NCD-related lifestyle risks among Bhutanese concerning the concept of happiness.

## 1. Introduction

Lifestyle changes may impact people's health and happiness [1–6]. Globally, the socioeconomic burden of non-communicable-diseases (NCDs) has been increasing [7–11]. According to a recent report by the WHO [12], the prevalence of diabetes is increasing in Bhutan in both genders, and 24.8% of the population is above the recommended weight level. Basic medical services are free and provided for all in Bhutan [13, 14]. Avoiding a future increase in medical expenses and social burdens is, therefore, an issue of highest priority, while the country has to simultaneously address unmet needs of infectious diseases, malnutrition, mother and child health, and so forth [15–17].

To emphasize sustainable development and the happiness of the people, Bhutan sets GNH (Gross National Happiness) as the basic philosophy of the national policy based on the Constitution [11]. The GNH consists of nine indicators, including living standard, education, cultural diversity, community vitality, time use, psychological wellbeing, ecological diversity, good governance, and health [1].

According to the definition of GNH philosophy, all of the nine indicators have the same weight, and their interactions affect each other. To measure health, including the measure of NCD prevention, we must consider health aspects and link these with other indicators, including psychological wellbeing, culture, and so on. The primary NCD prevention strategies require people to modify their daily lives, such as through a healthy diet and exercise [15–21]. It is, therefore, essential for Bhutan to understand how health is viewed, perceived, and practiced by the people concerning "happiness" or the GNH indicators. It is unclear whether NCD prevention strategies make people happy or unhappy. We cannot ignore the possibility that they might feel happier in their shorter lives or in living with NCDs without prevention compared to having strict health controls imposed upon them.

However, there is no research on health practice aspect and no analysis about relation to "happiness". Thus, this study qualitatively explores i) the current lifestyle-related risks for NCD, ii) socio-cultural factors that affect health-seeking behaviours, iii) the concept or value of happiness, and iv) information to inform future NCD prevention programs among urban and rural residents in southern Bhutan.

## 2. Methods

### 2.1. Research design

This is an exploratory qualitative study that combined in-depth interviews, participatory observation of daily life, and anthropometric measurements of steps, body mass index (BMI), abdominal girth, body fat, and blood pressure [22–25].

**Table 1. Topics of interview and observation.**

| |
|---|
| 1.Sociodemographic |
| 2.Medical history |
| 3.Health beliefs and behaviours |
| 4.Knowledge, recognition, and information resources about NCD |
| 5.Diet habits, daily physical activities |
| 6.GNH domains [8] |

## 2.2. Target population

Residents aged 18 years or older living in town (urban) and village (rural) in southern Bhutan.

## 2.3. Sampling

Purposive and theoretical sampling procedures were used. Participants were recruited through snowballing sampling process with multiple start points. Recruitments were terminated after thematic saturation was obtained [22–26].

## 2.4. Data collection and observation

Interviews and other observation were all performed by the PI (primary investigator, S-K.H) from June to September 2017, at the place determined by each participant, such as a private room at home or their workplace, paying careful attention to privacy protection. Semi-structured interviews (40–120 mins) were conducted according to the guides in Table 1, adjusting the order of, or omitting/adding questions considering the situation of each subject (S1 File). Due to ethnic diversity, English or local languages (Nepali, Dzongkha) were used in the interview with the cooperation of interpreters, who received prior training regarding the purpose, content, and study procedures. Participatory observation was performed taking field notes and photographs to observe daily activities of the participants (health behaviours, communication with others, food, cooking practice, and living and work environments) with special attention to the similarities with or the differences from the findings obtained in the interviews (S1 File). Furthermore, anthropometric measurements were conducted, including daily step counts for one week by pedometer, body mass index (BMI), abdominal girth, body fat, and blood pressure (S1 File).

## 2.5. Data analysis

Data were analysed according to the thematic approach shown in Table 2 [22–26]. Voice-recorded interviews were transcribed and translated into English by a bilingual co-investigator with back translation. To ensure credibility, certainty, and confirmability of the analysis, all

**Table 2. Procedures of thematic analysis.**

| |
|---|
| 1. Screening and familiarization with the data |
| 2. Combining interview and observation data |
| 3. Coding and categorizing |
| 4. Searching for themes |
| 5. Reviewing the themes |
| 6. Defining and naming the themes |

procedures were supervised by qualitative research experts and independently analysed by the researchers other than PI for researcher triangulation [22–26].

## 2.6. Ethical considerations

The Research Ethical Boards approved the study protocol from the Ministry of Health of the Royal Government of Bhutan (No. 2017–027) and the Kyoto University Faculty and Graduate School of Medicine, Ethics Committee (No. R1059). Potential participants were explained with purpose, schedule, and all procedures of the study and voluntary nature of participation and were included in the study only after obtaining written informed consent.

# 3. Results and discussion

## 3.1. Introduction of results

A total of 79 people participated. They were divided into four groups by the residential area (urban/rural) and age ($\geq$40/under). Participants in the urban area had higher educational and income levels and were more often self-employed or government officials, office workers, or housewives. In contrast, participants in the rural area were more often farmers. The daily step counts ranged from 4,500–5,500, the sleep hours were around seven, and body fat was 30–40% irrespective of the residential area or age. Participants having BMI $\geq$25 were higher in the urban area than the rural but exceeded 30% in all groups. Participants with high blood pressure were concentrated in the rural group aged 40 or more (41.7%) while being less than 10% in other groups, as shown in Table 3.

From the analysis, 41 codes and six themes emerged, as shown in Tables 4–9. Since common themes were found frequently, four groups were merged for final analysis.

Code validation is shown after each theme's explanation in Table. Due to the limited space, only part of the code validations is shown. Lines in "*italics*" are narratives from the participants with their age, residence, and participant number in the parenthesis. Lines that are not italicized are the observation from participatory observation or measurements. The questions from the interviewer are headed with "Q-."

## 3.2. Explanation of themes and codes

**Theme I: Rapid socio-economic changes (Table 4).** Both rural and urban areas have experienced getting out of poverty (Table 4, code No.1). Due to the development of hydroelectric power generation and factory zones, deforestation due to development and loss of farmland has occurred (Table 4, code No.2). Around 20 to 40 years ago, the inhabitants collected food from forests, kept cattle, and cultivated farm fields with cow dung.

However, they have experienced change from a self-sufficient life to a monetary society (Table 4, code No.3). Currently, most farmers grow money crops such as cardamom (a spice) and earn money and purchase foods such as Indian rice and daily necessities at the market. Additionally, due to changes in the natural environment, such as rivers drying up, paddies could not be maintained, and orange trees bore no fruit. Accordingly, the inhabitants had no choice but to make changes to their agricultural crops (Table 4, code No.4). Since the younger generation tends to leave the rural areas for the urban areas, this has caused the aging of agricultural workers and has increased the size of the service industry (Table 4, code No.5).

The inhabitants have experienced more convenient daily lives due to better infrastructure and the popularization of motors and home appliances (Table 4, code No.6). This has also led to improved physical access to processed goods, junk foods, alcohol, tobacco, and so forth (Table 4, code No.7). Not only physical access has improved, but also changes in access

**Table 3. Background characteristic of the participants.**

| | | Rural | | Urban | | Total |
|---|---|---|---|---|---|---|
| | | age ≥40 | age < 40 | age ≥40 | age <40 | |
| Age | | 56·3±10·8 | 28·1±4·9 | 49·7±5·0 | 32·4±4·8 | 41·2375±13·9 |
| Sex | | | | | | 79 |
| | -Men | 7 (8·86%) | 0 (0·00%) | 6 (7·59%) | 6 (7·59%) | 19 (24·05%) |
| | -Women | 16 (20·25%) | 17 (21·52%) | 7 (8·86%) | 20 (25·32%) | 60 (75·95%) |
| Religion | | | | | | 79 |
| | -Buddhist | 1 (1·27%) | 4 (5·06%) | 9 (11·39%) | 19 (24·05%) | 33 (41·77%) |
| | -Christian | 14 (17·72%) | 11 (13·92%) | 2 (2·53%) | 0 (0·00%) | 27 (34·18%) |
| | -Hindu | 9 (11·39%) | 1 (1·27%) | 2 (2·53%) | 7 (8·86%) | 19 (24·05%) |
| Education | | | | | | 79 |
| | -No education | 22 (27·85%) | 10 (12·66%) | 5 (6·33%) | 4 (5·06%) | 41 (51·90%) |
| | -< = class6 | 0 (0·00%) | 2 (2·53%) | 2 (2·53%) | 0 (0·00%) | 4 (5·06%) |
| | -< = class10 | 1 (1·27%) | 3 (3·80%) | 2 (2·53%) | 5 (6·33%) | 11 (13·92%) |
| | -< = class12 | 0 (0·00%) | 0 (0·00%) | 2 (2·53%) | 8 (10·13%) | 10 (12·66%) |
| | -< = Bachelors | 1 (1·27%) | 1 (1·27%) | 2 (2·53%) | 7 (8·86%) | 11 (13·92%) |
| | -Masters< = | 0 (0·00%) | 0 (0·00%) | 0 (0·00%) | 1 (1·27%) | 1 (1·27%) |
| | -Other | 0 (0·00%) | 0 (0·00%) | 0 (0·00%) | 1 (1·27%) | 1 (1·27%) |
| Annual Household Income (Nu) | | | | | | 79 |
| | <10,000 | 1 (1·27%) | 0 (0·00%) | 0 (0·00%) | 0 (0·00%) | 1 (1·27%) |
| | 10,000~49,999 | 6 (7·59%) | 3 (3·80%) | 0 (0·00%) | 0 (0·00%) | 9 (11·39%) |
| | 50,000~99,999 | 4 (5·06%) | 0 (0·00%) | 0 (0·00%) | 3 (3·80%) | 7 (8·86%) |
| | 100,000~199,999 | 6 (7·59%) | 7 (8·86%) | 2 (2·53%) | 0 (0·00%) | 15 (18·99%) |
| | 200,000~499,999 | 2 (2·53%) | 4 (5·06%) | 5 (6·33%) | 15 (18·99%) | 26 (32·91%) |
| | 500,000~ | 2 (2·53%) | 0 (0·00%) | 2 (2·53%) | 4 (5·06%) | 8 (10·13%) |
| | Nil | 3 (3·80%) | 2 (2·53%) | 4 (5·06%) | 4 (5·06%) | 13 (16·46%) |
| Numbers of family members | | 4·2±2·4 | 4·3±1·8 | 4·1±1·5 | 4·8±2·1 | 4·5±2·1 |
| Prevalence of Over Weight (%) | | 33·3% | 62·5% | 84·6% | 53·8% | |
| Daily Steps | | 5811±2558 | 4744±2167 | 7327±5218 | 5096±2247 | 5767±3272 |
| Sleep Hour | | 7·0±0·8 | 7·1±1·1 | 6·9±1·2 | 7·7±1·8 | 7·2±1·3 |
| Prevalence of High Blood Pressure | | 41·7% | 6·3% | 7·7% | 7·7% | |

∗Mean±SD or n(%), Over weight = (BMI ≧25), High Blood Pressure = (systolic ≧140mmHg or Diastolic ≧90mmHg) no matter using medicine or not.

to information due to the spread of the Internet, television, and education (Table 4, code No.8).

**Theme II: Situation of NCD-related risks (Table 5).** According to the increasing convenience of modern life, participant lifestyles showed the emergence of sedentary life (Table 5, code No.9). Their eating style has been affected by a preference for a high intake of fats, carbohydrates, salt, and spices (Table 5, code No.10) and a daily intake of junk food (Table 5, code No.11) was observed. Especially in the rural area, we found insufficient consultations or inadequate control of high blood pressure (Table 5, code No.12). Together with social changes, especially in the urban area, we found the emergence of new occupational-related stresses due to changes in labour conditions (Table 5, code No.13), and increased opportunities of eating out, such as eating out during office hours (Table 5, code No.14), drinking refreshments, and eating out as a form of leisure with friends or family. We also confirmed the occurrence of people being overweight in both urban and rural areas (Table 5, code No.15).

**Table 4. Theme I: Rapid socio-economic changes: Validation, code, categories and themes.**

| No | A part of validation | Code | Majority | Category |
|---|---|---|---|---|
| 1 | *"When we were kids, it was really difficult to earn even two ngultrums. We had to work the whole day just to earn two ngultrums or get two kilograms of millets."* (40≦age: rural) | Getting out of poverty | Whole | [1]. Rapid socio-economic and environmental changes |
| 2 | *"All the paddy fields have already been converted into commercial use. . .on our land, a mini hydropower plant was constructed, and we moved here. "*(40≦age: rural) | Deforestation due to development and loss of farmland | Rural 40≦ | |
| 3 | *"We used to eat rice. . . from our own paddy. But now I eat rice bought from the market."* (age<40: urban) | Change from a self-sufficient life to a monetary society | Urban, Rural 40≦ | |
| 4 | Q-When did you stop cultivating rice? *"It has been almost 40 years now."* Q-What do you sell from your farm? *"Ginger, butter, cheese, and cardamom are what we sell."* (40≦age: rural) | Changes in the types of crops (appearance of cash crops and change of natural environment) | Rural 40≦ | |
| 5 | Only elder people can make cheese in the family. Younger generations are engaged in factory work and shop management. | Aging of agricultural workers and the increase of the service industry | Rural 40≦ | |
| 6 | Taxis travel between rural and urban areas several times a day. We found traffic jams from taxis and private cars in town. | More convenient daily life due to better infrastructure and popularization of motors and home appliances | Whole | [2]. Lifestyle changes |
| 7 | *"(Starting to use rice-cooker and refrigerator from) five years ago, 2012. Little bit easy, and fast way to cook, and it's comfortable."* (age<40: urban) Stores were established in urban and rural areas. If people have money, it is very easy to get food and daily necessities such as junk food, soft drink, and alcohol. Children and adults use stores. | Improved physical access to processed goods, junk food, alcohol, tobacco, etc. | Whole | |
| 8 | *"When we were children, we cannot even watch TV. . . . (but now) We get mobiles, we can make calls."* (age<40: urban) | Changes in access to information due to the spread of the Internet, TV, and education | Whole | |

(Only part of the code validations is shown here. Lines in "*italics*" are narratives from the participants with their range, residence. Lines that are not italicized are the observation from participatory observation or measurements. The question from the interviewer as headed with "Q-.")

**Theme III: Current situation of health education about NCDs (Table 6).** Bhutan has experienced rapid social and economic changes, but an educational gap still exists across society. Medical services and education systems are improving, but NCD prevention through medical services is still inadequate (Table 6, code No.16). NCD patients, such as those with diabetes and high blood pressure, are receiving health guidance about nutrition and exercise. However, in practice, people cannot understand this because the information is insufficient and impractical (Table 6, code No.17) except for pregnant women, who are receiving sufficient nutrition guidance (Table 6, code No.18). School education provides insufficient basic education on nutrition, exercise, and NCDs (Table 6, code No.19).

**Theme IV: Knowledge, recognition, and preventing NCDs (Table 7).** Across every educational background and economic status, we found knowledge bias and misrecognition of nutrition, exercise, and NCDs (Table 7, code No.20). Participants recognize the need for healthy behaviours, but do not know how to practice them (Table 7, code No.21). Some highly educated people tried to search for health information online. However, most people lack basic knowledge and cannot recognize what information is accurate. Only people who have experienced sickness are trying to change their lifestyles dramatically (Table 7, code No.22). Without sickness, most participants experience difficulty in adopting healthier eating habits and exercise (Table 7, code No.23). We also found a gap between knowledge and practice (Table 7, code No.24).

**Theme V: Traditional customs and values (Table 8).** There is a traditional belief that values butter and cheese as being good for health (Table 8, code No.25). With improved access to processed foods, the trend of consuming foods with excess fat has been enhanced. Across both

**Table 5. Theme II: Situation of NCD-related risks: Variation, code, categories and themes.**

| No | A part of validation | Code | Majority | Category |
|---|---|---|---|---|
| 9 | The daily step counts ranged from 4,500–5,500 irrespective of age and residential area. "*(Usually I go to working place) by car, by my own car.*" (age<40: urban) | Emerging sedentary life | Whole | [3]. NCD-related risks |
| 10 | In cooking, participants usually put substantial amounts of oil into the pan to cook meats and vegetables and add a handful of salt. They also eat many foods as side dishes that are rich in butter and cheese. | Preference for high intake of fat, carbohydrates, salt, and spices | Whole | |
| 11 | In both rural and urban areas, we saw many children eating snacks often. "*(we take a soft drink) about eight or nine bottles in a week, maybe.*" (age<40: rural) | Daily intake of junk food | Part of Urban, Rural | |
| 12 | Hypertension was concentrated among people in rural areas aged ≥40 (41.7%) without appropriate medication. | Insufficient consultations or inadequate control of high blood pressure | Rural 40≦ | |
| 13 | A participant said, "*I have to correspond with over 100 phones per day,*" and showed me their mobile history. He checked the phone so many times during chatting, and he said: "*It is so stressful.*" (40≦age: urban) | The emergence of new occupational-related stress due to changes in labor conditions | Urban | |
| 14 | "*I like to go to restaurants. At least once per week, we go, sometimes with husbands, friends, sometimes the canteen in the office.…we can try different menus and enjoy them.*" (40≦age: urban) | Increased eating out | Urban | |
| 15 | Participants having a BMI ≧25 were higher in urban than rural areas but exceeded 30% in all groups. | The occurrence of being overweight | Part of Urban, Rural | |

(Only part of the code validations is shown here. Lines in "*italics*" are narratives from the participants with their range, residence. Lines that are not italicized are the observation from participatory observation or measurements. The question from the interviewer as headed with "Q-.")

rural and urban areas, a preference for chili and doma (Table 8, code No.26) and a habit of having dinner just before sleep was customary (Table 8, code No.27). We found a strong social cohesion and a hospitality mind-set. This helps people enhance their bonding, but it can be a cause of increased snacking and stress (Table 8, code No.28), as well as increased negative feelings from declining invitations of alcohol and tobacco (Tabl e8, code No.29).

Both rural and urban areas focus on social acceptance and find it difficult to limit doma, alcohol, and tobacco use at home (Table 8, code No.30). They treat their families well and make them a priority. Additionally, acceptance of the natural course of events is more valued than working too hard, enduring, or managing time to accomplish a goal (Table 8, code No.31).

We found that most participants found it challenging to change their behaviours, for example, they used to follow recommendations from religious leaders, such as vegetarianism and

**Table 6. Theme III: Current situation of health education: *Validation, code, categories and themes*.**

| No | A part of validation | Code | Majority | Category |
|---|---|---|---|---|
| 16 | Q-Have you learned the amounts of energy you take from foods in the training period? "*No idea……Everyone just advised me to take green leafy vegetables, foods that contain vitamins and minerals, etc.* Q-During health assistance education, did you learn about the nutritional components in vegetables and fruits? "*No, I don't remember about it.*" (age<40: urban) | Inadequate medical service | Whole | [4]. Medical services |
| 17 | "*I didn't do exercise. They (medical profession) told me, but they didn't teach this in action.*" (40≦age: rural) | Insufficient and impractical health guidance | Whole | |
| 18 | Q-Then have you heard about pregnancy nutrition? "*Yes.*" Q-What kind of things did you hear? "*Eat green vegetables and fruit that help to increase the blood production.*" (age<40: rural) | The spread of pregnancy term's nutrition guide | Below 40 | |
| 19 | *During my school time? Nutrition class? No… There is no class as nutrition class.* (age<40: urban) | School education with insufficient basic education on nutrition, exercise, NCD | Whole | [5]. School education |

(Only part of the code validations is shown here. Lines in "*italics*" are narratives from the participants with their range, residence. Lines that are not italicized are the observation from participatory observation or measurements. The question from the interviewer as headed with "Q-.")

**Table 7. Theme IV: Knowledge, recognition, and practice of prevention towards NCD: Validation, code, categories and themes.**

| No | A part of validation | Code | Majority | Category |
|---|---|---|---|---|
| 20 | *"We eat noodles a lot. . .. The baby also eats, and usually, we eat (noodles) for dinner. We try to have a light dinner. For my dad, we make roti. And we eat spaghetti and pasta. Eat light for dinner (for health)."* Q-Noodle, roti or pasta is light for you? *"Yes. Light food."* (age<40: urban) | Knowledge bias and misrecognition about nutrition, exercise, NCD | Whole | [6]. Fragmented knowledge and recognition of NCD |
| 21 | | Recognizes the need for healthy behaviors, but does not know healthy practices | Whole | |
| 22 | *"I took alcohol in the past as part of our culture, but once I got ill, I stopped drinking."* (40≦age: urban) | People who have experienced sickness are trying to change their lifestyle | Part of Urban, Rural | [7]. Factors promoting health behavior |
| 23 | *"Yes, I take doma a lot. Even the doctor told me to stop. . . I can't stop. I used to take 30–40 pieces a day."* (40≦age: urban) | Difficulty in adopting healthier eating habits and exercising | Part of Urban, Rural | [8]. Difficulty adopting healthy behaviors |
| 24 | A female participant said: *"Fatty food is not good for health."* However, during cooking time, she used much oil with plenty of local butter. | The gap between knowledge, recognition, and practice | Whole | |

(Only part of the code validations is shown here. Lines in "*italics*" are narratives from the participants with their range, residence. Lines that are not italicized are the observation from participatory observation or measurements. The question from the interviewer as headed with "Q-.")

**Table 8. Theme V: Traditional habits and value: Validation, code, categories and themes.**

| No | A part of validation | Code | Majority | Category |
|---|---|---|---|---|
| 25 | During cooking time, the participant said *"local butter and cheese is very good for your health,"* and she used plenty of local butter and cheese. (age<40: urban) | Value butter and cheese as good for health | Whole | [9]. Eating habits |
| 26 | *"For Bhutanese, we think salt and chili is important because, if there is no salt, even if we have put cheese and butter into everything, we know it will not be tasty."* (age<40: urban) | Preference for chili and doma | Whole | |
| 27 | Most participants eat dinner 1–2 hours before sleeping. | A habit of having dinner just before bedtime | Whole | |
| 28 | (A shop owner) Whenever her friends visit her shop, she said she serves tea and sweets for friends. | Increased snacking and stress caused by certain hospitality habits | Whole | [10]. Strong social cohesion |
| 29 | Q-Why did you start to take doma at that time? *"Just for pleasure and friends starting offering me, and I started."* Q-How about alcohol? *"Also, for pleasure, and occasionally if we don't take it, we feel backward."* (40≦age: rural) | Negative feelings from declining invitations of alcohol and tobacco | Whole | |
| 30 | Taking doma is a part of traditional Bhutanese culture. | Social acceptance: difficulty limiting doma, alcohol, and tobacco use at home | Part of Urban, Rural | [11]. Tolerance and social recognition |
| 31 | *"We don't have discipline. . .. We get lazy in the middle, after some times."* (age<40: urban)<br>They mentioned being "*selfish*" for managing their time, and they explained that "*here is stretch time.*" They do not try to follow the schedule too closely. | Acceptance of the natural course of events is more highly valued than working too hard, enduring, or managing time to accomplish a goal | Whole | |
| 32 | *"Lama advised me. . . to do prostration 10,000 times, so I started it every morning. . .I don't have any habit of exercise, but I will start. . .Bhutanese traditional form of yoga that is going to be taught by one of the Khenpo."* (age<40: urban) | Following recommendations from religious authorities: vegetarianism and prostration (gesture used in Buddhist practice) | Whole | [12]. Profound faith |
| 33 | *"I used to get sick, then everybody told me that I can get healed if I become Christian. . .. Even if I get sick now, I don't have to call a medium, but call a pastor and believer to pray. Till now I have not got sick again. . ."* (40≦age: rural) | Regardless of creed, deep faith that the meaning of life transcends the power of human beings | Whole | |
| 34 | *"To do good things helps in life after death."* (age<40: urban)<br>*"Only lama (religious authority) can give us how to prepare for death and how to spend our end-of-life."* (age<40: urban) | Religion takes a significant role in guiding end-of-life | Whole | |
| 35 | When a baby is sick in a village, some mothers used to visit Paw (who prays to cure illness using a tool). | Combination of traditional medicine, Western medicine, prayers, and indigenous spiritualism | Whole | |

(Only part of the code validations is shown here. Lines in "*italics*" are narratives from the participants with their range, residence. Lines that are not italicized are the observation from participatory observation or measurements. The question from the interviewer as headed with "Q-.")

**Table 9. Theme VI: Happiness: Validation, code, categories and themes.**

| No | A part of validation | Code | Majority | Category |
|---|---|---|---|---|
| 36 | *"Health is equally important for happiness."* (40≦age: rural) | Health is an important factor of happiness | Whole | [13]. Contribution of health to happiness |
| 37 | *"Money is not more important than our health. If I have good health, I can earn more money. But we can't buy health by money."* (age<40: rural)<br>*"The most important thing in my life is to be healthy, to keep my family happy and to go for praying."* (age<40: urban) | Health is more important for happiness than monetary enrichment | Whole | |
| 38 | *"My health, peace and harmony, wellbeing of all family (are important for happiness)."* (40≦age: rural)<br>*"If I get to eat and stay that is my happiness."* (40≦age: rural)<br>*"If I get to get eat delicious food and get to wear good cloths."* (40≦age: rural) | Family, house, food, clothes, peace, harmony, and religion are important for happiness | Whole | [14]. Other factors of happiness |
| 39 | Q-What factor determines your happiness? *"If the working environment is good, I feel happy. Then if the boss is good and even the salary, I feel happy."* (age<40: urban)<br>*"Factors. . .. . .i am free, independent, I don't have to depend for others."* (age<40: urban) | Work environment and working to be independent is important for happiness | Below 40 | |
| 40 | Q-Do you have some request for your child? *"Yes, do business. . .from where you can earn money."* (40≦age: rural)<br>*"Nowadays to get a job is not like our time. I would tell them to study hard and try to get a degree and job."* (40≦age: urban)<br>*"Study hard, something to do for this country."* (40≦age: urban) | Wish for their children to earn money and contribute to society | Whole | |
| 41 | *"To feel happy is important. . . .. Without happiness, we can't survive."* (age<40: rural)<br>*"No, I don't have any dreams. . .. So, I am happy with what I am (Laughs)."* (age<40: rural)<br>*"If I have no disease, how much I feel happy. If I can throw away diabetes and pressure, how much I feel happy. But even in current situation, I feel happy. . .."* (40≦age: urban) | Finding happiness in any situation is considered a virtue | Whole | [15]. Value of happiness |

(Only part of the code validations is shown here. Lines in "*italics*" are narratives from the participants with their age range, residence. Lines that are not italicized are the observation from participatory observation or measurements. The question from the interviewer as headed with "Q-.")

prostration (a gesture used in Buddhist practice) (Table 8, code No.32). Sometimes, acceptance of the natural course disrupts discipline and changes behaviour, at the same time sometimes their social bond and their belief helps them to become more disciplined or to change their behaviour. Through the experience of sickness, many changed their religion, but regardless of creed, there exists a deep faith that the meaning of life transcends the power of human beings (Table 8, code No.33). Moreover, religion takes a significant role in guiding end-of-life (Table 8, code No.34). The Bhutanese government established a medical system that consists of both traditional and Western medicine since people naturally use a combination of traditional medicine, Western medicine, prayers, and indigenous spiritualism (Table 8, code No.35).

**Theme VI: Happiness (Table 9).** Happiness is important in Bhutanese traditional values. For participants, health is an important element of happiness (Table 9, code No.36), and most people mentioned that health is more important for happiness than monetary enrichment (Table 9, code No.37). The participants recognized that family, houses, food, clothes, peace, harmony, and religion are important for happiness (Table 9, code No.38). Those participants under the age of 40 grasped that work environments and working to be independent is important for happiness (Table 9, code No.39). Participants over the age of 40 also wished for their children to earn money and contribute to society (Table 9, code No.40). However, regardless of their background, they recognized that "finding happiness in any situation is considered a virtue" (Table 9, code No.41). Moreover, valuing happiness helped them to live with diseases and be happy and accepting of their sadness or contradictions in their lives.

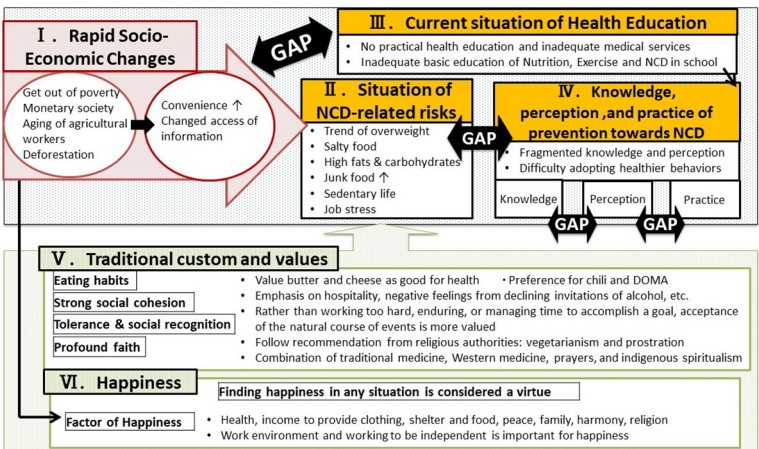

**Fig 1. Concept model.**

### 3.3. Summary of results

Four major gaps were identified (Fig 1) with the most significant gap being between rapid social and lifestyle changes and the situation of health education. Other gaps include the one between rising NCDs-related risk and people's perceptions, the gap between knowledge and perception and the gap between perception and practice. Rapid social changes were also found affecting people's health-seeking behaviours and the concept of happiness. Furthermore, traditional customs and values of happiness affect all situations negatively and positively. Moreover, the value of happiness helped them to live happily even with diseases and to accept their real lives.

## 4. Discussion

This is the first study which qualitatively examined the knowledge, perception, and practices related to NCDs among Bhutanese, particularly focusing on the philosophy of happiness, which is the fundamental value of Bhutanese culture reflected in the concept of GNH. This study revealed that rapid social and lifestyle changes are underway in Bhutanese society, creating knowledge, perception, and practice gaps that may put the Bhutanese society at risk of future NCD outbreaks.

Four major gaps were identified: 1) the gap between rapid lifestyle changes and the limited health education programs; 2) the gap between rising NCD-related risk and people's perceptions; 3) the gap between knowledge and perception; and 4) the gap between perception and practice. With these gaps, people are increasingly engaging in behaviours with a greater risk of NCDs, sometimes by a misunderstanding that they are having healthy behaviours or sometimes due to the lack of information on how to avoid the risks. Though many people understand the importance of reducing the intake of salt and fatty foods and the importance of exercise, irrespective of their education level, the majority did not know the number of calories of the foods they were consuming, the recommended level of daily calorie intake, and how to put such knowledge into practice.

We also found that the concept of happiness or traditional values and customs were associated with such situations in a complex manner. Traditional customs and values were found to affect the risk of NCDs through the belief that traditional foods (cheese and butter) are good for one's health, and through the "strong social cohesion and full hospitality" that makes it

difficult for people to decline smoking, drinking, or sweet foods even though they know they are not good for their health. Profound faith also affects their health behaviours positively or negatively. For example, when religion authorities give Bhutanese to do prostration, they will follow regardless heavy exercise or religion authorities give them some tips when they meet some difficulty. On the other hand, it may disturb their health if they give the people misguidance.

Concerning self-reported happiness, we found "health, family, house, food, clothes, peace, harmony, religion, work environment, and work for independence" to be important for happiness. Through participatory observations, we found that there was a possibility that factors of happiness, those which residents were unaware of, such as enough sleeping time, self-affirmation, social status, good governance, and good relationships with others. However, we cannot define and measure this through our study design.

However, we have gained a deeper understanding of the concept of happiness through this study, and we found it may also have mixed impacts on health. While it may promote health by underpinning the belief that health is essential for happiness and by helping policy makers to make health issue a policy priority, the well rooted cultural value of happiness of "finding happiness in any situation is considered a virtue" among Bhutanese may make them not to take any action for the health-related situation whatever it is. However, potentially positive impacts of this value of happiness may warrant special notion. We should note that, while some NCDs are preventable, not all NCDs and disabilities cannot eventually be prevented, especially under resource limited settings like Bhutan [8,10,14,15]. Therefore, while the importance of prevention is paramount, the importance of how to live with diseases, disorders, aging, and accepting death should not be underestimated. The Bhutanese value of happiness to accept living with diseases or disorders, aging, and death may help people to live with less stress with diseases or disabilities or even help them reduce social costs associated with such situations. Accordingly, given the increasing Bhutanese life expectancy along with the number of chronic diseases [15–17], we should consider how to harmonize the balance between health promotion and their traditional values and customs and the value of happiness.

As suggested in the emerging four gaps identified in our study, more efforts are urgently needed to balance between health promotion and the traditional values and customs and the value of happiness. Based on these observations and respecting their harmony with society [1], four points are recommended. 1) There is a need for more practical health education: Although most people understand the importance of reducing the intake of salt and fatty foods and the importance of exercise, because of the lack of education programs, people do not know how to put such knowledges into practice [27]. It is necessary to communicate knowledge in practical forms in health education programs. 2) For health education group or community-based approach rather than individual approach should be used [28–31]: We found a strong social cohesion and a hospitality mind-set among Bhutanese. While this helps people enhance their bonding, but it can be a cause of increased snacking and stress as well as increased negative feelings from declining invitations of alcohol and tobacco. To solve this problem, group or community-based approaches rather than individual tutorials may be more effective to create new norms good for health in such as working place, or village, religious, and generational communities, and so forth. 3) Education of basic knowledge relating to nutrition and exercise in primary and junior high school curricula and the dissemination of awareness to the parents of children should be promoted: Primary and junior high school is a life period that establishes lifestyles [29]. Moreover, for a culture that values families, it is possible to encourage effective behavioural changes by disseminating awareness to parents and residential communities through children [29–42]. 4) Collaboration with the religious authorities is needed: The people's deep faith and devotion could help make difficult changes in behaviour.

Religion in Bhutanese society is closely related to the concept of happiness. "Collaboration" with religious authorities and empowering people's healthy behaviour is important. Such an approach maintains peace of mind and relieves new sources of social stress.

## 5. Limitation

This study holds several limitations. Since this is a qualitative study with a limited sample size, the findings will not be generalized unless they are combined with population-based quantitative research. However, we could get rich information for each participant, and we collected data and analysed till we could achieve theoretical saturation. Possible biases should also be considered in both interviews and analysis, however this study closely worked with a bilingual co-investigator and assistants and analyses were carefully triangulated. Moreover, using this study finding, we will continue the next phase of research to contribute to health and happiness.

## 6. Conclusion

This study revealed several important gaps in NCDs among Bhutanese society; the gap between changing society/lifestyles and health education, the gap between actual NCD risk and its perception, and the gap between perception and practice. The study also revealed that the traditional values and customs and the value of happiness had effects on the risk of NCDs. From these findings, it is recommended that community-based and more practical comprehensive health promotion programs for NCD prevention involving the community and religious leaders will have positive impacts in preventing NCDs in Bhutan.

## Supporting information

**S1 File.**
(DOCX)

## Acknowledgments

We would like to thank Editage (www.editage.com) for English language editing.

We appreciate the collaboration of all participants, assistants, and the Khesar Gyalpo University of Medical Sciences of Bhutan, and the Ministry of Health of Bhutan, as well as all the supports extended from the School of Public Health and the Graduate School of Asian-African Studies, Kyoto University, Japan.

## Author Contributions

**Conceptualization:** Hiromi Kohori-Segawa, Patou Masika Musumari, Sonia Pilar Sugimoto Watanabe, Masako Ono-Kihara, Masahiro Kihara.

**Data curation:** Hiromi Kohori-Segawa, Chimi Dema, Yankha Dorji, Masahiro Kihara.

**Formal analysis:** Hiromi Kohori-Segawa, Masako Ono-Kihara, Masahiro Kihara.

**Funding acquisition:** Hiromi Kohori-Segawa, Teeranee Techasrivichien, Ryota Sakamoto, Masahiro Kihara, Yuichi Imanaka.

**Investigation:** Hiromi Kohori-Segawa.

**Methodology:** Hiromi Kohori-Segawa, Patou Masika Musumari, Ryota Sakamoto, Masako Ono-Kihara, Masahiro Kihara.

**Project administration:** Hiromi Kohori-Segawa, Kunzang Dorji, Ugyen Wangdi, Chimi Dema, Yankha Dorji, Masahiro Kihara.

**Resources:** Hiromi Kohori-Segawa.

**Supervision:** Chencho Dorji, Ryota Sakamoto, Masako Ono-Kihara, Masahiro Kihara, Yuichi Imanaka.

**Validation:** Hiromi Kohori-Segawa, Kunzang Dorji, Ugyen Wangdi, Chimi Dema, Yankha Dorji, Patou Masika Musumari, Sonia Pilar Sugimoto Watanabe, Ryota Sakamoto, Masako Ono-Kihara, Masahiro Kihara.

**Visualization:** Hiromi Kohori-Segawa, Masahiro Kihara.

**Writing – original draft:** Hiromi Kohori-Segawa.

**Writing – review & editing:** Hiromi Kohori-Segawa, Kunzang Dorji, Ugyen Wangdi, Patou Masika Musumari, Teeranee Techasrivichien, Sonia Pilar Sugimoto Watanabe, Ryota Sakamoto, Masahiro Kihara, Yuichi Imanaka.

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
