## [Decision Letter · Decision Letter 0]

17 Mar 2020

PONE-D-20-01072

A qualitative study on knowledge, perception, and practice related to non-communicable diseases in relation to happiness among rural and urban residents in Bhutan

PLOS ONE

Dear Dr Hiromi,

Thank you for submitting your manuscript to PLOS ONE. After careful consideration, we feel that it has merit but does not fully meet PLOS ONE’s publication criteria as it currently stands. Therefore, we invite you to submit a revised version of the manuscript that addresses the points raised during the review process.

We would appreciate receiving your revised manuscript by 16th April 2020. To enhance the reproducibility of your results, we recommend that if applicable you deposit your laboratory protocols in protocols.io, where a protocol can be assigned its own identifier (DOI) such that it can be cited independently in the future. For instructions see: http://journals.plos.org/plosone/s/submission-guidelines#loc-laboratory-protocols

We look forward to receiving your revised manuscript.

Kind regards,

Russell Kabir, PhD

Academic Editor

PLOS ONE

Journal Requirements:

2. Please include additional information regarding the interview guide or script used in the study and ensure that you have provided sufficient details that others could replicate the analyses. For instance, if you developed a guide as part of this study and it is not under a copyright more restrictive than CC-BY, please include a copy, in both the original language and English, as Supporting Information.

4. We note you have included a table to which you do not refer in the text of your manuscript. Please ensure that you refer to Table 5, 6, 7, 8 in your text; if accepted, production will need this reference to link the reader to the Table.

Reviewers' comments:

Reviewer's Responses to Questions

**Comments to the Author**

1. Is the manuscript technically sound, and do the data support the conclusions?

Reviewer #1: Yes

Reviewer #2: Partly

2. Has the statistical analysis been performed appropriately and rigorously? 

Reviewer #1: N/A

Reviewer #2: Yes

3. Have the authors made all data underlying the findings in their manuscript fully available?

Reviewer #1: Yes

Reviewer #2: Yes

4. Is the manuscript presented in an intelligible fashion and written in standard English?

Reviewer #1: Yes

Reviewer #2: Yes

5. Review Comments to the Author

Reviewer #1: The manuscript should be revised linking how psychological well-being is part of happiness and why it matters for health through knowledge about NCDs. Also, the manuscript should refocus on health seeking behaviour, especially in context of NCDs.

Reviewer #2: The paper undoubtedly presents a very good initiative to consider happiness as an indicator of NCD prevalence in Bhutan albeit happiness is an abstract term and difficult to measure. The study undertook qualitative analysis while only IDI and observation study cannot define the state of happiness completely. A mixed methods approach could be excellent in this case to measure their property and land ownership and occupational status for which happiness state might vary. Also, there could be some potential confounders that needed to be considered in this study like, domestic violence, neighbourhood relations and most importantly subjective social status, i.e., occupational hierarchy that also evidently influence individuals' happiness and well-being. Also, there can be bias regarding the happiness related statements by the respondents. However, the authors did great by considering the traditional behaviour in this study on which mental satisfaction and happiness depend to a good extent.

Ultimately, this paper needs to discuss about other potential factors that have significant influence in determining happy or unhappy individuality. Furthermore, inclusion of quantitative analysis could provide more credible findings.

6. PLOS authors have the option to publish the peer review history of their article (what does this mean?). If published, this will include your full peer review and any attached files.

Reviewer #1: No

Reviewer #2: No

---

## [Author Response · Author response to Decision Letter 0]

16 May 2020

Dear Editor and Reviewers:

I am very appreciative of your review and comments. I am pleased to submit a revised version of our article for publication in the Journal of Plos One, titled “A qualitative study on knowledge, perception, and practice related to non-communicable diseases in relation to happiness among rural and urban residents in Bhutan”. The article was authored by Hiromi Kohori-Segawa, Chencho Dorji, Ugyen Wangdi, Kunzang Dorji, Chimi Dema, Yankha Dorji, Patou Masika Musumari, Teeranee Techasrivichien, Sonia Pilar Sugimoto Watanabe, Ryota Sakamoto, Masako Ono-Kihara, Masahiro Kihara, and Yuichi Imanaka.

We modified the following six points according to your review and comments.

1. File naming

2. Supplements (Interview guide and so on)

3. Data Availability statement

Data Availability: This study was conducted in the Kingdom of Bhutan which, is a small Himalayan country. The transcript, anthropometric, and participatory observation data and photos, contain participant information including disease history, relationships with others, religion, family history, social status, and sensitive issues, such as family problems and mental health. Given these factors, the data cannot be made publicly available considering the potential breach of privacy and confidentiality. However, the authors can share the data if requested through the Research Ethics Board of Health at the Royal Government of Bhutan, and the ethics committee of Kyoto University.

Contact information: 

The Research Ethics Board of Health, Royal Government of Bhutan, PABX: + 975-2-322602, 322351, 328091, 328092, 328093 (Extension 333) Fax: 324649 (Research No. 2017-027)

Ethics Committee of Kyoto University Graduate School and Faculty of Medicine, Kyoto University Hospital, E-mail: ethcom@kuhp.kyoto-u.ac.jp (Research Number: No. R1059)

4. Links with code on tables in our manuscript

5. The introduction was modified according to reviewer 1’s comments (Line 80 to 88 in manuscript)

6. The discussion and limitations were modified according to reviewer 2’s comments (Line 310 to 317, 359 to 360)

All authors have read and approved the manuscript. It has not been published in any form and is not under consideration for publication elsewhere.

Thank you for your kind consideration.

Sincerely, 

Hiromi Kohori-Segawa, MPH.

---

## [Editor Report · Decision Letter 1]

22 May 2020

A qualitative study on knowledge, perception, and practice related to non-communicable diseases in relation to happiness among rural and urban residents in Bhutan

PONE-D-20-01072R1

Dear Dr. Hiromi,

We are pleased to inform you that your manuscript has been judged scientifically suitable for publication and will be formally accepted for publication once it complies with all outstanding technical requirements.

With kind regards,

Russell Kabir, PhD

Academic Editor

PLOS ONE
---

## [Editor Report · Acceptance letter]

15 Jun 2020

PONE-D-20-01072R1 

A qualitative study on knowledge, perception, and practice related to non-communicable diseases in relation to happiness among rural and urban residents in Bhutan 

Dear Dr. Segawa:

I'm pleased to inform you that your manuscript has been deemed suitable for publication in PLOS ONE. Congratulations! Your manuscript is now with our production department. 

Kind regards, 

on behalf of

Dr. Russell Kabir 

Academic Editor

PLOS ONE